# Trading Off Resource Budgets For Improved Regret Bounds

**Damon Falck**[*]
University of Oxford
damon.falck@gmail.com

**Thomas Orton**[*]
University of Oxford
thomas.orton@cs.ox.ac.uk

## Abstract

In this work we consider a variant of adversarial online learning where in each round one picks $B$ out of $N$ arms and incurs cost equal to the *minimum* of the costs of each arm chosen. We propose an algorithm called Follow the Perturbed Multiple Leaders (FPML) for this problem, which we show (by adapting the techniques of Kalai and Vempala [2005]) achieves expected regret $\mathcal{O}(T^{\frac{1}{B+1}} \ln(N)^{\frac{B}{B+1}})$ over time horizon $T$ relative to the *single* best arm in hindsight. This introduces a trade-off between the budget $B$ and the single-best-arm regret, and we proceed to investigate several applications of this trade-off. First, we observe that algorithms which use standard regret minimizers as subroutines can sometimes be adapted by replacing these subroutines with FPML, and we use this to generalize existing algorithms for Online Submodular Function Maximization [Streeter and Golovin, 2008] in both the full feedback and semi-bandit feedback settings. Next, we empirically evaluate our new algorithms on an online black-box hyperparameter optimization problem. Finally, we show how FPML can lead to new algorithms for Linear Programming which require stronger oracles at the benefit of fewer oracle calls.

## 1 Introduction

Adversarial online learning is a well-studied framework for sequential decision making with numerous applications. In each round $t = 1, \ldots, T$, an adversary chooses a hidden cost function $c_t : \mathcal{A} \to [0, 1]$ from a set of arms $\mathcal{A}$ to costs in $[0, 1]$. An algorithm must then choose an arm $a_t \in \mathcal{A}$, and incurs cost $c_t(a_t)$. In the *full feedback* setting (Online Learning with Experts (OLwE)), the algorithm then observes the cost function $c_t$. In the *partial feedback* setting (Multi-Armed Bandits (MAB)) the algorithm only observes the incurred cost $c_t(a_t)$. The objective is to find algorithms which minimize *regret*, defined as the difference between the algorithm's cumulative cost and the cumulative cost of the single best arm in hindsight.

In this paper we consider a search-like variant of these problems where in each round one can pick a *subset* of arms $S_t \subset \mathcal{A}$ with $|S_t| = B \geq 1$, and *keep the arm with the smallest cost*. This variant appears naturally in many settings, including:

1. Online algorithm portfolios [Gomes and Selman, 2001]: In each round $t$, one receives a problem instance $x_t$, and can pick a subset $S_t \subset \mathcal{A}$ of algorithms to run in parallel to solve $x_t$. For example, $x_t$ could be a boolean satisfiability (SAT) problem, and $\mathcal{A}$ could be a collection of different SAT solving algorithms. We let $c_t(a) = 0$ if $a$ solves $x_t$ and $c_t(a) = 1$ otherwise. Then if any $a \in S_t$ finds a solution to $x_t$ we incur 0 cost in this round. Another example is online hyperparameter optimization (see Section 4).

---

[*]Equal contribution.

2. Online bidding [Chen et al., 2016a]: In each round $t$, an auctioneer sets up a first-price auction for bidders $S_t \subset \mathcal{A}$. Each bidder $a \in \mathcal{A}$ has a price $1 - c_t(a)$ they are willing to pay, and the auctioneer receives $\max_{a \in S_t} 1 - c_t(a) = 1 - \min_{a \in S_t} c_t(a)$, and so maximizing revenue is equivalent to minimizing costs.

3. Adaptive network routing [Awerbuch and Kleinberg, 2008]: In each round $t$, a network router receives a data packet $x_t$ and can pick a selection of network routes $S_t \subset \mathcal{A}$ to send it to its destination via in parallel. Let the cost $c_t(a)$ of a route $a$ be the total time taken for $x_t$ to reach its destination via $a$; the router receives cost $\min_{a \in S_t} c_t(a)$ equal to the smallest delay.

In many applications the budget $B$ is a restricted resource (e.g. compute time or number of cores) we would like to keep small; this paper studies how one can trade off budget resources for better guarantees on the standard regret objective.

Formally, for any randomized algorithm **ALG** which chooses subset $S_t \subset \mathcal{A}$ in round $t$, and thus incurs cost $c_t(S_t) := \min_{a \in S_t} c_t(a)$, define

$$R_T^*(\textbf{ALG}) := \max_{c_1, \ldots, c_T} \mathbb{E}\left[ \sum_{t=1}^{T} c_t(S_t) - \min_{a^* \in \mathcal{A}} \sum_{t=1}^{T} c_t(a^*) \right]$$

to be the worst-case expected regret of **ALG** relative to the single best arm in hindsight, where the expectation is with respect to the randomness of **ALG**.[*] What guarantees can we give on $R_T^*$ as a function of our budget $B$? In the full feedback setting when $B = 1$, this is the standard OLwE problem where it is known that $R_T^* = \Omega(\sqrt{T})$ and there are algorithms which achieve $R_T^* \leq 2\sqrt{T \ln(N)}$ [Lattimore and Szepesvári, 2020], where $N = |\mathcal{A}|$. When $B = N$ the algorithm which chooses $S_t = \mathcal{A}$ in each round achieves $R_T^* = 0$. But what bounds on $R_T^*$ can be achieved in the intermediate regime when $1 < B < N$? To the best of our knowledge this question has not been directly answered by any prior work.

## 1.1 Contributions

**Theoretical results:** We present a new algorithm for this learning problem called *Follow the Perturbed Multiple Leaders* (**FPML**), and show that in the full feedback setting $R_T^*(\textbf{FPML}) \leq \mathcal{O}(T^{\frac{1}{B+1}} \ln(N)^{\frac{B}{B+1}})$. This allows for a direct trade-off between the budget $B$ and the regret bound (in particular, allowing resources $B \geq \Omega(\ln(T))$ leads to regret *constant* in $T$) and recovers the familiar $\mathcal{O}(\sqrt{T \ln(N)})$ bound when $B = 1$. We then show that in the *semi-bandit feedback* setting (where the algorithm finds out only the costs of the arms it chooses) this bound can be converted to $R_T^*(\textbf{FPML}) \leq \mathcal{O}(T^{\frac{1}{B+1}}(K \ln(N))^{\frac{B}{B+1}})$ if one has unbiased cost estimators bounded in $[0, K]$.

We also consider the more general problem of Online Submodular Function Maximization (OSFM), for which prior work gives an online greedy algorithm **OG** [Streeter and Golovin, 2008]. When given a budget of $B$ per round, **OG** achieves regret $\mathcal{O}(\sqrt{TB \ln(N)})$ with respect to $(1 - e^{-1})\text{OPT}(B)$, where $\text{OPT}(B)$ is the performance of the best fixed length-$B$ schedule (see Section 3 for a formal definition of OSFM). Note that in this guarantee the regret benchmark is a function of the algorithm budget. By replacing a subroutine in **OG** with **FPML**, we generalize **OG** to a new algorithm **OG**$_{\textbf{hybrid}}$. Unlike **OG**, **OG**$_{\textbf{hybrid}}$ is able to give regret bounds against benchmarks which are decoupled from the algorithm budget. This allows one to more easily quantify the trade-off of increasing the budget against a fixed regret objective. As a special case, we are able to show that having a budget of $B = B'\lceil \ln(T)^2 \rceil$ per round allows one to achieve regret $\mathcal{O}(B' \ln(T) \ln(N))$ with respect to $\text{OPT}(B')$. One interpretation of this result is that if you are willing to increase your budget (e.g. runtime) by a factor of $\ln(T)^2$, you are able to improve your performance guarantee benchmark from $(1 - e^{-1})\text{OPT}(B')$ to $\text{OPT}(B')$. Likewise, your regret growth rate in terms of the number of rounds changes from $\mathcal{O}(\sqrt{T})$ to $\mathcal{O}(\ln(T))$.

Finally, in Section 5 we show how to use **FTML** to generalize a technique for solving linear programs assuming access to an oracle which solves relaxed forms of the linear program. To obtain an $\varepsilon$-approximate solution to the linear program requires $\left(\frac{1}{\varepsilon}\right)^{\frac{B+1}{B}} (4\rho)^{\frac{B+1}{B}} (1 + \ln(n))$ oracle calls, where

---

[*]Here we consider an oblivious adversary model for simplicity, but we believe the results of this paper carry through to adaptive adversaries as well.

the parameters $(B, \rho)$ are related to the power of the oracle and $n$ is the number of linear constraints. The case $B = 1$ coincides with known results.

**Experimental results:**  We benchmark both **FPML** and **OG$_{\textbf{hybrid}}$** on an online black-box hyper-parameter optimization problem based on the 2020 NeurIPS BBO challenge [Turner et al., 2021]. We find that both these new algorithms outperform **OG** for various compute budgets. We are able to explain why this happens for this specific dataset, and discuss the scenarios under which each algorithm would perform better.

**Techniques:**  Minimizing $R_T^*$ is an important subroutine for a large variety of applications including Linear Programming, Boosting, and solving zero sum games [Arora et al., 2012]. Traditionally an experts algorithm such as **Hedge** [Littlestone and Warmuth, 1994], which pulls a single arm per round, will be used as a subroutine to minimize $R_T^*$. We highlight how in the cases of OSFM and Linear Programming, one can simply replace a single arm $R_T^*$-minimizing subroutine with **FPML** and get performance bounds with little or no alteration to the original proofs. The resulting algorithms have improved bounds (due to improved bounds on $R_T^*$ when $B > 1$) at the cost of qualitatively changing the algorithm (e.g. requiring a larger budget or more powerful oracle). This is significant because it highlights the potential of how bounds on $R_T^*$ when $B > 1$ can lead to new results in other application areas. In Section 2.1 we also highlight how the proof techniques of Kalai and Vempala [2005] for bounding $R_T^*$ in the traditional experts setting can naturally be generalized to the case when $B > 1$, which is of independent interest.

## 1.2   Relation to prior work

One can alternatively formulate more gewe consider as receiving the maximum *reward* $r_t(a) = 1 - c_t(a)$ of each arm chosen instead of the minimum cost. In this maximum of rewards formulation, the problem fits within the OSFM framework where (a) all actions are unit-time and (b) the submodular *job* function is always a maximum of rewards. The rewards formulation of the problem has also been separately studied as the K-MAX problem (here $K = B$) Chen et al. [2016a]. In the OSFM setting, Streeter and Golovin [2008] give an online greedy approximation algorithm which guarantees $\mathbb{E}[(1 - e^{-1})\mathrm{OPT}(B) - \mathrm{Reward}_T] \leq \mathcal{O}(\sqrt{TB\ln(N)})$ in the full feedback adversarial setting, where $\mathrm{OPT}(B)$ is the cumulative reward of the best fixed subset of $B$ arms in hindsight, and $\mathrm{Reward}_T$ is the cumulative reward of the algorithm. A similar bound of $\mathcal{O}(B\sqrt{TN\ln(N)})$ can be given in a semi-feedback setting. Conversely in the full feedback setting, Streeter and Golovin [2007] shows that any algorithm has worst-case regret $\mathbb{E}[\mathrm{OPT}(B) - \mathrm{Reward}_T] \geq \Omega(\sqrt{TB\ln(N/B)})$ when one receives the maximum of rewards in each round. Chen et al. [2016a] study the K-MAX problem and other non-linear reward functions in the stochastic combinatorial multi-armed bandit setting. Assuming the rewards satisfy certain distributional assumptions, they give an algorithm which achieves distribution-independent regret bounds of $\mathbb{E}[(1 - \varepsilon)\mathrm{OPT}(B) - \mathrm{Reward}_T] \leq \mathcal{O}(\sqrt{TBN\ln(T)})$ for $\varepsilon > 0$ with semi-bandit feedback. Note however that we consider the adversarial setting in this paper.

More broadly, these problems fall within the combinatorial online learning setting where an algorithm may pull a subset of arms in each round. Much prior work has focused on combinatorial bandits where the reward is linear in the subset of arms chosen, which can model applications including online advertising and online shortest paths [Cesa-Bianchi and Lugosi, 2012, Audibert et al., 2014, Combes et al., 2015]. The case of non-linear reward is comparatively less studied, but having non-linear rewards (such as max) allows one to model a wider variety of problems including online expected utility maximization [Li and Deshpande, 2011, Chen et al., 2016a]. As some examples of prior work in the stochastic setting, [Gopalan et al., 2014] uses Thompson Sampling to deal with non-linear rewards of functions of subsets of arms (including the max function), but requires the rewards to come from a known parametric distribution. Chen et al. [2016b] considers a model where the subset of arms pulled is randomized based on pulling a 'super-arm', and the reward is a non-linear function of the values of the arms pulled. In the adversarial setting, Han et al. [2021] study the combinatorial MAB problem when rewards can be expressed as a $d$-degree polynomial.

In contrast to prior work which focuses on giving algorithms which compete against benchmarks which have *the same* budget as the algorithm, this work is concerned with the trade-off between regret bounds and budget size. We focus on giving regret bounds against $\mathrm{OPT}(1)$, and we use this result in Section 3 to get regret bounds against $\mathrm{OPT}(B')$ for $B' < B$ in OSFM. Decoupling the regret

benchmark $\text{OPT}(B')$ from the algorithm budget $B$ can be useful when one would like to control the strength of a regret bound against a specific target $\text{OPT}(B')$ for theoretical or applied reasons. For example Arora et al. [2012] survey a wide variety of applications which rely on bounding $R_T^*$, but bounds such as $\mathbb{E}[(1 - e^{-1})\text{OPT}(B) - \text{Reward}_T] \leq \mathcal{O}(\sqrt{TB\ln(N)})$ do not immediately imply useful bounds on $\text{OPT}(1) - \text{Reward}_T$.

## 2 Follow the Perturbed Multiple Leaders

We begin by considering the full feedback setting. We first check that allowing the algorithm to choose $B > 1$ arms per round, while only competing against the best *single* fixed arm in hindsight, does not make the problem trivial. We do this by showing that any deterministic algorithm with budget $B < N$ still achieves linear regret in the number of rounds. This is achieved by setting $c_t(a) = 1$ if $a \in S_t$, $c_t(a) = 0$ otherwise.

**Proposition 1.** *In the full feedback setting, any deterministic algorithm with arm budget $B \leq N$ per round has worst-case regret $R_T^* \geq \left(1 - \frac{B}{N}\right)T$.*

Likewise, it can be shown that the algorithm which chooses a uniformly random subset of $B$ arms in each round has worst-case expected regret at least $T(1 - \frac{B}{N})^B$ (achieved by having one arm have cost 0 across all rounds and every other arm having cost 1). These two observations show that any solution for achieving sub-linear regret in $T$ requires randomization which depends in some non-trivial way on the prior observed costs even when $B > 1$.

### 2.1 Generalizing Follow the Perturbed Leader

Choosing the current lowest perturbed-cost arm in each round, *Follow the Perturbed Leader* (**FPL**) [Kalai and Vempala, 2005], is a well-known regret minimization technique which achieves optimal worst-case regret against adaptive adversaries in the OLwE setting. In this section we generalize the **FPL** algorithm to *Follow the Perturbed Multiple Leaders* (**FPML**). In each round, **FPML** perturbs the cumulative costs of each arm by adding noise, and then picks the $B$ arms with lowest cumulative perturbed cost. This is precisely **FPL** when $B = 1$. We show how one can extend the proof techniques of Kalai and Vempala [2005] in a natural way to prove that **FPML** achieves worst-case regret $R_T^*(\textbf{FPML}) \leq 2T^{\frac{1}{B+1}}(1 + \ln(N))^{\frac{B}{B+1}}$.

---

**Algorithm 1 FPML$(B, \varepsilon)$**

**Require:** $N \geq B \geq 1, \varepsilon > 0$.
    Initialize the cumulative cost $C_0(a) \leftarrow 0$ for each arm $a \in \mathcal{A}$.
    **for** round $t = 1, \ldots, T$ **do**
        1. For each arm $a \in \mathcal{A}$, draw a noise perturbation $p_t(a) \sim \frac{1}{\varepsilon}$ Exp.
        2. Calculate the perturbed cumulative costs for round $t - 1$, $\tilde{C}_{t-1}(a) \leftarrow C_{t-1}(a) - p_t(a)$.
        3. Pull the $B$ arms with the lowest perturbed cumulative costs according to $\tilde{C}_{t-1}$. Break ties arbitrarily.
        4. Update the cumulative costs for each arm, $C_t(a) \leftarrow C_{t-1}(a) + c_t(a)$.
    **end for**

---

**Theorem 2.** *In the full feedback setting, where $S_t \subset \mathcal{A}$ is the subset of arms chosen by **FPML** in round $t$, we have:*

$$\max_{c_1, \ldots, c_T} \mathbb{E}\left[(1 - \varepsilon^B)\sum_{t=1}^{T} c_t(S_t) - \min_{a^* \in \mathcal{A}}\sum_{t=1}^{T} c_t(a^*)\right] \leq \frac{(1 + \ln(N))}{\varepsilon}.$$

*In particular, for $\varepsilon = ((\ln(N) + 1)/T)^{1/(B+1)}$, we have*

$$R_T^*(\textbf{FPML}) \leq 2T^{\frac{1}{B+1}}(1 + \ln(N))^{\frac{B}{B+1}}.$$

The proof follows the same three high level steps which appear in Kalai and Vempala [2005] for **FPL**, but we extend these ideas to the case where $B > 1$. We first observe that the algorithm which picks the $B$ lowest cumulative cost arms in each round only incurs regret when the best arm in round $t$ is not one of the best $B$ arms in round $t - 1$.

**Lemma 3.** *Consider a fixed sequence of cost functions $c_1, \ldots, c_T$. Let $a_t^{*,j}$ be the $j^{th}$ lowest cumulative cost arm in hindsight after the first $t$ rounds, breaking ties arbitrarily. Let $S_t^* := \{a_{t-1}^{*,j} \mid j \in [B]\}$ be the set of the $B$ lowest cost arms at the end of round $t-1$. Then for each $i \in [T]$,*

$$R_i := \sum_{t=1}^{i} c_t(S_t^*) - \min_{a^* \in \mathcal{A}} \sum_{t=1}^{i} c_t(a^*) \leq \sum_{t=1}^{i} \mathbb{1}[a_t^{*,1} \notin S_t^*]$$

*and $R_i - R_{i-1} \leq \mathbb{1}[a_i^{*,1} \notin S_i^*]$.*

This is a generalization of the familiar result that when $B = 1$, following the leader has regret bounded by the number of times the leader is overtaken [Kalai and Vempala, 2005].

The second step is to argue that if the cumulative costs are perturbed slightly, it becomes unlikely that the event $\{a_t^{*,1} \notin S_t^*\}$ will occur. One way to see this is as follows: fix a round $t$, and let $C_{t-1}(a)$ be the cumulative cost of $a$ at the end of round $t-1$. Let $M := C_{t-1}(a^{*,B+1})$. Then every $a \in S_t^*$ has $C_{t-1}(a) \leq M$. If it is also true that $C_{t-1}(a) < M - c_t(a)$ for any $a \in S_t^*$ then the event $\{a_t^{*,1} \notin S_t^*\}$ cannot occur. This is because $C_t(a) < (M - c_t(a)) + c_t(a) = M$ but any $a' \in \mathcal{A} - S_t$ has $C_t(a') \geq M$, so $a' \neq a_t^{*,1}$. If we had initially perturbed each $C_{t-1}(a)$ by subtracting independent exponential noise $p(a) \sim \frac{1}{\varepsilon}\mathrm{Exp}$, then conditional on $M$ the event $\{C_{t-1}(a) < M - c_t(a)\}$ is jointly independent for each $a \in S_t^*$. Moreover the probability of this inequality not holding is equal to $\mathbb{P}[p(a) < v + c_t(a)|p(a) \geq v]$ for $v$ equal to the unperturbed cost of $a$ at round $t-1$ minus $M$, which is bounded by $\varepsilon c_t(a)$ (due to the memorylessness property of the exponential distribution).

**Lemma 4.** *Fix a sequence of cost functions $c_1, \ldots, c_T$. Let $C_i(a) = \sum_{t=1}^{i} c_t(a)$ and $\tilde{C}_i(a) = C_i(a) - p(a)$ be the perturbed cumulative cost of arm $a$ at the end of round $i$, where $p(a) \sim \frac{1}{\varepsilon}\mathrm{Exp}$. Let $\tilde{a}_t^{*,j}$ be the $j^{th}$ lowest cumulative cost arm in hindsight after the first $t$ rounds using these perturbed costs, and let $\tilde{S}_t^* = \{\tilde{a}_{t-1}^{*,j} \mid j \in [B]\}$. Then*

$$\mathbb{E}\left[\mathbb{1}\left[\tilde{a}_t^{*,1} \notin \tilde{S}_t^*\right]\right] \leq \mathbb{E}\left[\varepsilon^B c_t(\tilde{S}_t^*)\right].$$

Again, when $B = 1$ this argument and bound coincides with the argument given by [Kalai and Vempala [2005]].

The final step is to combine Lemmas 3 and 4 to argue that **FPML** achieves expected regret at most $\mathbb{E}\left[\varepsilon^B \sum_{t=1}^{T} c_t(\tilde{S}_t^*)\right]$ with respect to the perturbed cumulative cost $\tilde{C}_T$. Since $\max_{a \in \mathcal{A}} \mathbb{E}[p(a)] \leq \frac{1+\ln(N)}{\varepsilon}$ we can argue we also achieve low expected regret with respect to the unperturbed cost $C_T$. In the setting of this paper, drawing new random perturbations $p_t(a)$ in each round is not strictly necessary (we can take $p_t(a) = p_1(a)$ for $t > 1$), but it is necessary to achieve regret bounds when cost functions can depend on prior arm choices of the algorithm (the *adaptive* adversarial setting). In the setting of this paper where the costs are fixed, the expected regret in either case is the same.

**Probabilistic guarantees:** One advantage of this proof technique is that the regret is bounded using the positive random variable $\sum_{t=1}^{T} \mathbb{1}\left[\tilde{a}_t^{*,1} \notin \tilde{S}_t^*\right]$. This means that one can apply methods like Markov inequality to give a probabilistic guarantee of small regret, which is substantially stronger than saying the regret is small in expectation.

**Comments on settings of parameters:** When $B = 1$ we recover the standard $\mathcal{O}(\sqrt{T \ln(N)})$ regret bound for the OLwE problem. For $B > 1$, the regret growth rate as a function of the number of rounds is $T^{\frac{1}{B+1}}$. In particular, when $B = \Omega(\ln(T))$ grows slowly with the number of rounds, the expected regret becomes $\mathcal{O}(\ln(N))$ and does not grow with the number of rounds $T$. If we use a tighter inequality in the proof of Lemma 4, it is possible to get constant expected regret when $B = \ln(T)\ln(A)$ grows slowly with the number of arms and rounds.

**Lower bounds:** A standard technique for constructing lower bounds in the online experts setting with $B = 1$ is to consider costs which are i.i.d. Bernoulli$(p)$ [Lattimore and Szepesvári, 2020]. Unfortunately this technique fails when $B > 1$ because the expected cost of the minimum of $B$ i.i.d. Bernoulli random variables is generally smaller than the expected cost of the best arm in hindsight

unless $p$ is very close to 1. We are able to show very weak lower bounds in the full feedback setting of $R_T^* = \Omega(\ln(N))$ for constant $B$ and $T = \Omega(\ln(N))$, but there is a substantial gap between this and the upper bound. We think constructing stronger lower bounds is an interesting problem for future work which may require new analysis techniques.

**Partial feedback:**  The *semi-bandit feedback* setting is a form of partial feedback where the algorithm only observes the individual costs of the arms it pulls. It can be shown that passing unbiased cost function estimates to **FPML** results in a similar regret bound in the semi-bandit feedback setting; the result is specific to **FPML** and using unbiased cost functions does not generally work for any $R_T^*$-minimizing algorithm when $B > 1$ because of the non-linearity of the cost function. In the case of **FPML** this is not an issue because the same bounding technique using Lemma 3 holds in expectation when using unbiased cost estimators. A naïve way to generate unbiased cost estimates in this setting is to use an additional arm to uniformly sample costs; in Section 4 we explore geometric sampling [Neu and Bartók, 2013] for getting unbiased cost estimates which is effective in practice.

**Proposition 5.** *Define the algorithm* **FPML-partial** *which simulates* **FPML**, *passing it unbiased cost estimates $\hat{c}_t \in [0, K]$ at round $t \in [T]$,[\*] and copies the arm choices of* **FPML**. *Then we have*

$R_T^*(\textbf{FPML-partial}) \leq \ln(N)/\varepsilon + T(1 - \mathrm{e}^{-K\varepsilon})^B$. *For $\varepsilon = \left(\frac{\ln(N)}{TK^B}\right)^{\frac{1}{B+1}}$, $R_T^*(\textbf{FPML-partial}) \leq \mathcal{O}(T^{\frac{1}{B+1}}(K\ln(N))^{\frac{B}{B+1}})$.*

## 3 Generalized regret bounds for Online Submodular Function Maximization

Streeter and Golovin [2008] considered the more general problem of Online Submodular Function Maximization which captures a number of previously studied problems as special cases. For the sake of emphasizing the key ideas, we restrict attention to the full feedback setting where each action has unit duration. The OSFM problem in this setting is as follows:

**Definition 1.** *Define a* schedule *to be a finite sequence of actions[\*] $a \in \mathcal{A}$, and let $\mathcal{S}$ be the set of all schedules; the* length $\ell(S)$ *of a schedule $S \in \mathcal{S}$ is the number of actions it contains. Define a* job *to be a function $f : \mathcal{S} \to [0, 1]$ such that for any schedules $S_1, S_2 \in \mathcal{S}$ and any action $a \in \mathcal{A}$:*

1. *$f(S_1) \leq f(S_1 \oplus S_2)$ and $f(S_2) \leq f(S_1 \oplus S_2)$* **(monotonicity)**;

2. *$f(S_1 \oplus S_2 \oplus \langle a \rangle) - f(S_1 \oplus S_2) \leq f(S_1 \oplus \langle a \rangle) - f(S_1)$* **(submodularity)**.

**Definition 2** (Online Submodular Function Maximization). *The problem consists of a game with $T$ rounds. We are given some fixed $B > 0$ and at each round $t \in [T]$ we must choose a schedule $S_t \in \mathcal{S}$ with $\ell(S_t) \leq B$ to be evaluated by a job $f_t$ which is only revealed after our choice. The goal is to maximize the cumulative output $\mathrm{Reward}_T := \sum_{t=1}^{T} f_t(S_t)$.*

Streeter and Golovin [2008] propose an online greedy algorithm **OG** which achieves the guarantee $(1 - e^{-1})\mathrm{OPT}(B) - \mathrm{Reward}_T \leq \mathcal{O}(\sqrt{TB\ln(N)})$ in expectation, where $\mathrm{OPT}(B)$ is the cumulative reward of the best fixed schedule of length $B$ in hindsight. In this section we explain how to use **FPML** to extend their algorithm to allow a trade-off between budget resources and regret bounds.

The algorithm **OG** has two key ideas. Suppose that we start with an empty schedule $S_0 := \emptyset$. The first idea is that if we knew $f_1, \ldots, f_T$ in advance, we could greedily construct $S_i := S_{i-1} \oplus \langle \arg\max_{a_i \in \mathcal{A}} \sum_{t=1}^{T} f_t(S_{i-1} \oplus \langle a_i \rangle) - f_t(S_{i-1}) \rangle$ for $i = 1, \ldots, B$, where $a_i$ is the best greedy arm in hindsight for greedy round $i$. It can be shown that submodularity then implies $(1 - e^{-1})\mathrm{OPT}(B) \leq \sum_{t=1}^{T} f(S_B)$. Since we don't know $f_1, \ldots, f_T$ in advance, the second idea is to run $B$ copies of a $R_T^*$ regret minimizer, where the $i^{\text{th}}$ copy tries to compete with achieving the same *improvement* of cumulative reward as the best fixed greedy action $a_i$ in hindsight. The regret bound on the $i^{\text{th}}$ copy with respect to the best greedy arm in hindsight is $\mathcal{O}(\sqrt{T\ln N})$; across the $B$ copies one can show the net regret compared to the offline greedy solution is bounded by $\mathcal{O}(\sqrt{TB\ln N})$, which is where

---

[\*]More formally: $\mathbb{E}[\hat{c}_t(\cdot) \mid \mathcal{F}_{t-1}] = c_t(\cdot)$ where $\mathcal{F}_{t-1}$ is the $\sigma$-algebra generated by all the randomness up to and including round $t - 1$.

[\*]In their original problem definition actions may each have a different associated *duration*.

the final bound comes from. In summary, **OG** works as follows: for each round $t \in [T]$, run $B$ greedy rounds. In greedy round $i = 1, \ldots, B$, pull the arm $a_i^t$ proposed by the $i^{\text{th}}$ black box $R_T^*$-minimizer, set $S_{t,i} := S_{t,i-1} \oplus \langle a_i^t \rangle$, and feed back the greedy rewards $r_{t,i}(a) = f_t(S_{t,i-1} \oplus \langle a \rangle) - f_t(S_{t,i-1})$ to the $i^{\text{th}}$ black box.

We propose a hybrid version of **OG**, called **OG**$_{\textbf{hybrid}}$, which for any budget $B$ allows us to compete asymptotically well against $\text{OPT}(B')$ for any chosen $B' \leq B/\ln(T)$. The algorithm **OG**$_{\textbf{hybrid}}$ is based on the following two changes to **OG**:

1. Instead of having $B$ greedy rounds, we have $B' \ln(T)$ greedy rounds. One can show that the extra factor of $\ln(T)$ allows one to drop the $(1 - e^{-1})$ term in the regret bound.

2. Instead of running a one-arm-pulling $R_T^*$ minimizer in each greedy round, we run **FPML** which pulls $\lfloor B/B' \ln(T) \rfloor$ arms. This allows us to improve the regret bound for each of the $B' \ln(T)$ $R_T^*$-minimizers and directly translates to a tighter overall regret bound.

Besides these two changes, the algorithm is identical to that in Streeter and Golovin [2008].

More generally, let **OG**$_{\textbf{hybrid}}(B, \tilde{B})$ denote the algorithm where each **FPML** box has a budget of $\tilde{B}$ and there are $B/\tilde{B}$ greedy rounds, so that the total number of arms pulled in each round is $B$. This algorithm is a hybrid of **OG** and **FPML** in the sense that **OG**$_{\textbf{hybrid}}(B, 1)$ is **OG** with *Follow the Perturbed Leader* as the $R_T^*$-minimizing subroutine. On the other hand, **OG**$_{\textbf{hybrid}}(B, B)$ is **FPML**. Varying $\tilde{B}$ allows us to interpolate between these two algorithms by varying the budget we give to the **FPML** subroutines.

On a technical note, because we are pulling $\tilde{B} \geq 1$ arms for each greedy choice, we require a slight strengthening on the monotonicity condition which is common to many practical applications including the experiments we consider in the next section:

**Assumption 1.** *In addition to monotonicity and submodularity, each job $f : \mathcal{S} \to [0, 1]$ also satisfies $f(S_1 \oplus S_2 \oplus S_3) \geq f(S_1 \oplus S_3)$ for any schedules $S_1, S_2, S_3 \in \mathcal{S}$.*

We can then give the following bounds:

**Theorem 6.** *For any choices of $B', \tilde{B}$, under Assumption 1 and with budget $B := \lceil B' \widetilde{B} \ln(T) \rceil$, algorithm **OG**$_{\textbf{hybrid}}(B, \tilde{B})$ experiences expected regret*

$$\mathcal{O}\left( B' \ln(T) T^{1/(\widetilde{B}+1)} \ln(N)^{\widetilde{B}/(\widetilde{B}+1)} \right)$$

*relative to the best-in-hindsight fixed schedule of length $B'$. In particular, if $\widetilde{B} = \lceil \ln(T) \rceil$ (and so $B = B' \lceil \ln(T) \rceil^2$) the regret is bounded by $\mathcal{O}(B' \ln(T) \ln(N))$.*

This bound allows us the flexibility of trading off a schedule budget $B$ for how tightly we would like to compete with the best fixed schedule of length $B' \leq B$ in hindsight.

**Partial feedback:** Streeter and Golovin [2008] extend their algorithm to handle partial feedback by replacing the $R_T^*$ regret minimizers with the bandit algorithm **Exp3** [Auer et al., 2002] which only requires feedback on the arms which are pulled. Likewise, one can replace **FPML** with **FPML-partial** in **OG**$_{\textbf{hybrid}}$ to get an algorithm which gives regret bounds in a semi-bandit feedback setting (where we can observe $f_t(S)$ for any schedule $S$ consisting of actions which were pulled in round $t$). We empirically compare the partial feedback versions of **OG** and **OG**$_{\textbf{hybrid}}$ in the next section.

## 4   Experiments: online hyperparameter optimization

The problem of *black-box optimization*—where a hidden function is to be minimized using as few evaluations as possible—has recently generated increased interest in the context of hyperparameter selection in deep learning [Snoek et al., 2012, Liu et al., 2020b, Bouthillier and Varoquaux, 2020]. As a result, the 2020 NeurIPS BBO Challenge [Turner et al., 2021] invited participants' optimizers to compete to find the best possible configurations of several ML models on a number of common datasets, given a limited budget of training cycles for each. One of the key findings was that sophisticated new algorithms are normally outperformed on average by techniques that ensemble

**Table 1:** Experimental results for $B = 6$ and $B = 4$.

| $B = 6$ | | |
|---|---|---|
| Algorithm | Mean Reward | StD |
| **BIH**$(6)$ | 0.901 | NA |
| **FPML-partial**$(6)$ | 0.888 | 0.0072 |
| **OG$_{\text{hybrid}}$**$(6, 3)$ | 0.836 | 0.0111 |
| **OG$_{\text{hybrid}}$**$(6, 2)$ | 0.814 | 0.0143 |
| **OG$_{\text{hybrid}}$**$(6, 1)$ | 0.785 | 0.0137 |
| **OG** | 0.767 | 0.0157 |
| $B = 4$ | | |
| Algorithm | Mean Reward | StD |
| **BIH**$(4)$ | 0.836 | NA |
| **FPML-partial**$(4)$ | 0.813 | 0.0108 |
| **OG$_{\text{hybrid}}$**$(4, 2)$ | 0.756 | 0.0149 |
| **OG$_{\text{hybrid}}$**$(4, 1)$ | 0.716 | 0.0178 |
| **OG**$(4)$ | 0.689 | 0.0151 |

existing methods [Liu et al., 2020a], i.e. optimizers tend to have varying strengths and weaknesses that are suited to different task types. In a scenario where many hyperparameter selection problems are to be processed (e.g. in a data center) and limited computing resources are available, it may thus be desirable to learn over time how best to choose $B$ optimizers to apply independently to each problem (e.g. to run in parallel on $B$ available CPU cores). This is a natural partial feedback application of bandit algorithms that pull multiple arms per round and receive the best score found across the optimizers which are run.

### 4.1 Experimental setup

We follow a similar approach to the NeurIPS BBO Challenge; the optimization problem at each round $t \in [T]$ is to choose the hyperparameters of either a multi-layer perceptron (MLP) or a lasso classifier for one of 184 classification tasks from the Pembroke Machine Learning Benchmark [Olson et al., 2017] (so $T = 368$). At each round the bandit algorithm must select $B$ Bayesian black-box optimizers from a choice of 9 to run in parallel on the current problem; the received *reward*[*] for each optimizer was calculated as a $[0, 1]$-normalized measure of where the best training loss attained sits between **(a)** the expected best loss from a random hyperparameter search and **(b)** an estimate of the best possible loss attainable (the same approach used by the Bayesmark package [Uber, 2020]). Rewards are only observed for the optimizers which are run (semi-bandit feedback). For each budget level $B = 1, \ldots, 6$, we ran a benchmarking study comparing the performance of the following bandit algorithms in this setting: (1) **FPML-partial**$(B)$ from Section 2.1, using geometric sampling [Neu and Bartók, 2013] to construct unbiased cost estimates (see appendix for details). (2) **OG$_{\text{hybrid}}$**$(B, \tilde{B})$ from Section 3, with **FPML-partial** as a subroutine. We do this for varying values of $\tilde{B}$ to see how performance changes. (3) **OG**$(B)$, the partial feedback version of the original online greedy algorithm from Streeter and Golovin [2008]. We benchmark performance against **BIH**$(B)$, the score of the Best fixed subset of size $B$ In Hindsight. In all cases the $\varepsilon$ parameter for the bandit subroutines **FPML-partial** and **Exp3** was set to their theoretically optimal values given $B$, $N$ and $T$ without any fine-tuning (for **FPML-partial** we use Proposition 5). We run each algorithm setting 100 times to estimate the mean and standard deviation of the performance.

### 4.2 Results and discussion

As expected, **OG$_{\text{hybrid}}$**$(B, 1)$ has very similar performance to **OG**$(B)$ in all cases because they implement essentially the same algorithm (the difference being due to different choices of one-arm pulling $R_T^*$ minimizers, **FPML-partial**$(1)$ and **Exp3**). However, we notice that in every instance,

---

[*]Here we use rewards to be consistent with the setting of Online Submodular Function Maximization.

allocating more budget to the **FPML-partial** subroutine in **OG$_{\text{hybrid}}$** (increasing $\tilde{B}$) while keeping the overall budget constant improved the average score performance. This means that **OG$_{\text{hybrid}}$** was always at least as good as **OG** for all parameter settings, and that **FPML-partial** outperformed both of these algorithms in all cases. This is perhaps surprising, because **OG** and **OG$_{\text{hybrid}}$** are designed to achieve low regret against the stronger benchmark of the best subset of $B' \geq 1$ arms in hindsight, while **FPML-partial** is only designed to achieve low regret with respect to the single best optimizer in hindsight. Towards explaining this observation, we notice that for this particular dataset, the best subset of $B$ arms in hindsight happens to be very similar to the set of the individual best performing $B$ arms in hindsight, and **FPML** achieves low regret with with respect to the latter (see Proposition 7 below). This raises the interesting question of when certain regret objectives might be better than others in practice.

**Proposition 7.** *In the full feedback setting, the expected regret of **FPML** with $B$ arms relative to the set of the individual best performing $B$ arms in hindsight is at most $[1 - B^{-1} + \ln(|\mathcal{A}|/B)]/\varepsilon + T(1 - \exp(-\varepsilon B))$.*

**Synthetic tasks:** We also evaluated these partial feedback algorithms in a number of synthetic environments, exploring examples where **(a)** the optimal subset of $B$ arms, **(b)** the subset of $B$ arms chosen greedily, **(c)** the individual best performing $B$ arms, perform in various ways relative to each other; **OG** approximates **(b)** and **FPML-partial** approximates **(c)**, so the closeness of either of these to **(a)** determines each algorithm's performance. We find that problem instances exist where **OG** outperforms **FPML-partial** and vice versa. See the appendix for details.

## 5   Connections to other problems

Arora et al. [2012] surveys a wide variety of algorithmic problems and shows how they can all be solved using a $R_T^*$ minimizing subroutine in the standard OLwE setting with $B = 1$. The purpose of this section is to highlight the relative ease with which algorithms like **FPML** can sometimes be *plugged in* to existing algorithms which use an $R_T^*$ minimizer as black box subroutine, with little or no alteration to their proofs of correctness. We already saw an example of this in Section 3, where the regret minimizing subroutine was replaced with **FPML**. The resulting algorithm gave improved regret guarantees at the cost of higher budget requirements, without changing the structure of the original proof given in Streeter and Golovin [2008]. In this section we take Linear Programming as an example, and illustrate what its plug-in algorithm looks like. We are unsure whether the resulting algorithm is necessarily useful because it requires a more powerful oracle than the one supposed in Arora et al. [2012]; but we do think that exploring the algorithms which result from this plug-in technique more broadly may be an area of interest for future work.

### 5.1   Linear Programming

We consider the Linear Programming (LP) problem from Arora et al. [2012]: Given a convex set $P$, an $n \times m$ matrix A with entries in $\mathbb{R}$, and a vector $b \in \mathbb{R}^n$, the task is to find an $x \in P$ such that $Ax \geq b$, or determine that no such $x$ exists. We assume that we have an oracle which solves the following easier problem (where $A_i$ denotes the $i^{\text{th}}$ row of $A$):

**Definition 3.** *A $(\rho, B)$-bounded oracle for $\rho \geq 0$ is an algorithm, which when given a joint distribution $d$ over $[n]^B$, finds an $x \in P$ such that*

$$\mathbb{E}_{(i_1,\ldots,i_B) \sim d} \left[ \min_{i \in \{i_1,\ldots,i_B\}} A_i x - b_i \right] \geq 0$$

*or determines that no such $x$ exists. If an $x$ is found, then $\forall i \in [n], |A_i x - b_i| \leq \rho$.*

When $B = 1$ this is a simplified version of the oracle defined in Arora et al. [2012], and it is equivalent to an oracle which can find $x \in P$ which satisfies a single linear constraint $d^\top A x \geq d^\top b$ (when viewing $d$ as a vector in $\mathbb{R}^n$). When $B = n$, solving this problem can be as hard as solving the original LP problem by jointly choosing $i_j$ to have probability mass 1 on the $j^{\text{th}}$ linear constraint. $1 < B < n$ represents an intermediate regime where the oracle needs to find an $x$ which 'fools' the joint distribution $d$. For simplicity, we assume that we have access to an oracle which takes as input

bounded cost functions $c_1, \ldots, c_{t-1}$, and outputs the joint distribution $d_t$ over arms of **FPML** in round $t$ after observing cost functions $c_1, \ldots, c_{t-1}$. In practice such an oracle could be achieved by e.g. sampling arm draws from **FPML** to approximate $d_t$.

**Proposition 8.** *Let $\varepsilon > 0$. Suppose there exists a $(\rho, B)$-bounded oracle for the feasibility problem $\exists x \in P$ s.t. $Ax \geq b$. Then there is an algorithm which either finds an $x \in P$ s.t. $\forall i \in [n], A_i x \geq b_i - \varepsilon$, or correctly concludes that the problem is infeasible. The algorithm makes at most $T = \left(\frac{1}{\varepsilon}\right)^{\frac{B+1}{B}} (4\rho)^{\frac{B+1}{B}} (1 + \ln(n))$ calls to the $(\rho, B)$-bounded oracle and **FPML** oracle, with a total runtime of $\mathcal{O}(T)$.*

The proof technique for general $B \geq 1$ is essentially identical to the proof given in Arora et al. [2012] for $B = 1$ except that we are able to use a stronger bound on $R_T^*$ (see appendix for details).

**Comment on bound:** When $B = 1$, we require $\Omega((\frac{1}{\varepsilon})^2)$ steps in order to find an $x$ which is $\varepsilon$-close to satisfying the constraint. The $(\frac{1}{\varepsilon})^2$ term comes from the fact that when $B = 1$, the average regret for OLwE is $\Omega(\sqrt{T}/T) = T^{-\frac{1}{2}}$, so it takes $T \geq (\frac{1}{\varepsilon})^2$ steps for the average regret to be $\leq \varepsilon$. The quadratic dependence on an accuracy parameter is therefore common in many applications which use OLwE with $B = 1$ as a subroutine (including Boosting [Schapire, 1990] and solving zero sum games [Freund and Schapire, 1999]). For general $B \geq 1$, we only require $\mathcal{O}((\frac{1}{\varepsilon})^{\frac{B+1}{B}})$ steps (suppressing terms related to $n$ and $\rho$) for the average regret to be $\leq \varepsilon$. In the case of Linear Programming, this is at the expense of requiring a stronger oracle for the problem.

## 6 Conclusion

This paper presented a new algorithm, Follow the Perturbed Multiple Leaders, which allows one to directly trade off budget constraints for bounds on regret. We showed how **FPML** can be used as a subroutine to generate new algorithms for Online Submodular Function Optimization and Linear Programming which trade off resources and oracle power for improved performance guarantees.

We also highlight a number of areas for future work: (a) **Lower bounds:** Can we reduce the gap between the upper and lower bounds of this problem? Improving lower bounds may require new techniques than are traditionally used in the OLwE setting. (b) **Plug-in algorithms:** Are there other cases where using **FPML** in existing algorithms can lead to new theoretical results? And when are these new algorithms practically useful? (c) **Which regret benchmarks are useful in practice:** The experiments of Section 4 showed that algorithms designed to minimize regret with respect to a single arm can sometimes in practice outperform algorithms designed to minimize regret with respect to the stronger benchmark of the best subset of arms. Are certain regret objectives better than others in different practical applications?

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
