# OpenReview forum: "Trading Off Resource Budgets For Improved Regret Bounds"
_NeurIPS.cc/2022/Conference — NeurIPS 2022 Accept_

### Official Review · Reviewer_SYiQ · 2022-07-05

**Rating:** 5
**Confidence:** 4
**Soundness:** 3 good
**Presentation:** 4 excellent
**Contribution:** 2 fair

**Summary:**

This paper studies adversarial online learning problems where the algorithm chooses a subset of actions of size $B$ (instead of a single action) at each round and incurs the minimum cost among the chosen actions. The goal is to study how this additional `budget' per round leads to better regret bounds for the problem. The authors propose an extension of the FTPL algorithm, called Follow the Top perturbed Leaders, and obtain regret bounds for both full information and semi-bandit feedback settings. Finally, it has been shown how the proposed algorithm can be used as a sub-routine in state-of-the-art algorithms for other problems such as online submodular maximization and linear programming to obtain budget-dependent bounds without changing the previous proofs too much.

**Questions:**

- Explain the challenges faced when extending the proof of FTPL to the more general setting with a budget of $B$.
- Explain how the online submodular maximization problem fits the framework introduced in this paper.
- The distribution Exp has been used several times in the paper without a precise definition.
- The notation $\bigoplus$ is used in Section 3 without a precise definition.
- The name FTL used in the paper is a bit confusing. FTL usually refers to the Follow The Leader algorithm. However, in this paper, FTL refers to the extension of the Follow The Perturbed Leader (FTPL) algorithm.

**Limitations:**

Yes, the authors have clearly addressed the limitations of their work. The paper is mostly theoretical and does not need a discussion regarding potential negative societal impacts.

**Strengths And Weaknesses:**

Strengths:
- Comprehensive set of numerical experiments to verify the theoretical findings.
- The problem framework is well-motivated via several real-world applications and is quite interesting.
- Proof sketches provided in the text are quite insightful and provide high-level ideas of the proof without going into details.

Weaknesses:
- I'm not sure how the idea of choosing $B$ actions per round applies to the online submodular maximization problem. In online submodular maximization, the goal is to pick a subset of cardinality at most $B$ at each round to maximize the overall utility. Given that the rewards functions are monotone, it is natural to compare the performance of the algorithm against a fixed benchmark set with the same cardinality constraint $B$. That's why obtaining regret bounds against benchmark actions with smaller cardinality constraints (i.e., Theorem 6) is not reasonable. Also, the reward at round $t\in[T]$ for a chosen action $S_t$ ($|S_t|\leq B$) is $f_t(S_t)$, not the $\max_{a \in S_t}f_t(a)$ reward considered in this paper.
- The algorithm seems like a straightforward extension of the FTPL algorithm to the framework with a budget of $B$ and It is not clear to me how challenging this generalization is. The authors need to highlight the challenges and discuss how they overcame them to obtain regret bounds for the more general setting.

---

> ### Author Response · Authors · 2022-07-31
> **Response**
>
> We thank the reviewer for their constructive comments and for taking the time to review this paper. We hope the following will address their questions, and possibly persuade them to raise their score.
>
> > Explain the challenges faced when extending the proof of FTPL to the more general setting with a budget of $B$.
>
> A key challenge was in developing a proof technique which could solve the problem introduced in this paper. The fact that the analysis technique which ultimately worked could be interpreted as a natural generalisation of FPL was an unintentional and pleasant surprise. We spent a considerable amount of time trying different techniques which did not work. This was challenging because:
>
> 1. To the best of our knowledge, there do not exist any prior results for proving regret bounds when the algorithm has a budget different from the benchmark. There were therefore no prior references on how to approach this problem.
>
> 2. Many of the standard ideas for bounding regret in the combinatorial bandits setting fail when the cost function is non-linear. The non-linear case is considerably less well understood than the linear case.
>
> 3. It was not obvious that approaching the problem through the lens of Following the Perturbed Leader (FPL) would work. There are multiple analysis techniques for getting regret bounds for FPL in the standard setting, but only one of these seemed useful in the setting of this paper.
>
> > Explain how the online submodular maximization problem fits the framework introduced in this paper.
>
> Thank you for raising this question. We agree that the paper should make this relationship more explicit, and we will add this to the introduction of the paper. Reviewer Gn74 asked a similar question, so we will refer to that response to avoid duplication.
>
> > It is natural to compare the performance of the algorithm against a fixed benchmark set with the same cardinality constraint $B$. That's why obtaining regret bounds against benchmark actions with smaller cardinality constraints (i.e., Theorem 6) is not reasonable.
>
> If one has a budget of $B$, we agree that considering regret bounds against $\mathrm{OPT}(B)$ is one natural measure of performance. However, giving regret bounds against $\mathrm{OPT}(B)$ for this problem is challenging; prior work only gives bounds against $(1-e^{-1})\mathrm{OPT}(B)$ (L226). In comparison, this paper gives bound against  $\mathrm{OPT}(B)$ if you are willing to increase your budget by a factor of $\ln(T)^2$ (Theorem 6).
>
> Most importantly, the purpose of Theorem 6 in this paper is to be able to quantify the relative performance improvement as the budget is increased while the benchmark is held fixed. This is perhaps best illustrated by a direct comparison: The bounds from prior work (Streeter and Golovin [2008], L226) give the regret guarantee
> $$R_{T}=\max_{f_1,\dots,f_t} \mathop{\mathbb{E}}\left[(1-e^{-1})\mathrm{OPT}(B)-\sum_{t=1}^{T} f(S_t)\right] \leq \mathcal{O}(\sqrt{TB \ln(N)})$$
> when one has a budget of $B$. Theorem 6 says that if we are willing to increase our budget by the logarithmic factor $\ln(T)^2$, we are able achieve the guarantee $$R_{T}=\max_{f_1,\dots,f_t} \mathop{\mathbb{E}}\left[\mathrm{OPT}(B)-\sum_{t=1}^{T} f(S_t)\right] \leq \mathcal{O}(B\ln(T)\ln(N)).$$
>
> For example, in the case of online hyperparameter optimization, this means that if you are willing to increase your runtime (budget) by a factor of $(\ln T)^2$, you can improve your hyperparameter optimizer score guarantee against $\mathrm{OPT}(B)$ by a factor of $(1-e^{-1})$ and have the additive regret term grow like $\mathcal{O}(\ln T)$ instead of $\mathcal{O}(\sqrt{T})$. This gives a clear quantification of the marginal value of increasing the budget. On the other hand, if we tried to use the bounds of Streeter and Golovin [2008] to quantify how performance improves when we increase the budget by a factor of $\ln(T)^2$, we would get
> $$R_{T}=\max_{f_1,\dots,f_t} \mathop{\mathbb{E}}\left[(1-e^{-1})\mathrm{OPT}(B(\ln(T))^2)-\sum_{t=1}^{T} f(S_t)\right] \leq \mathcal{O}(\sqrt{TB \ln(T)^2 \ln(N)}).$$
> But this cannot be used to quantify the relative improvement: the benchmark has changed from $(1-e^{-1})\mathrm{OPT}(B)$ to $(1-e^{-1})\mathrm{OPT}(B\ln(T)^2)$ (which cannot be directly compared) and the additive regret term has deteriorated from $\mathcal{O}(\sqrt{TB\ln(N)})$ to $\mathcal{O}(\sqrt{TB \ln(T)^2 \ln(N)})$.
>
> ### Definition of Exp and $\oplus$
>
> Thank you for raising this. By $\frac{1}{\epsilon}\mathrm{Exp}$ we mean an exponential random variable with probability density function $\epsilon e^{-\epsilon x}$. $S_1 \oplus S_2$ is the concatenation of two ordered action schedules $S_1,S_2$. We will add this information to the paper.
>
> ### Naming of Algorithm 1
>
> Thank you for suggesting a name change. We agree that calling Algorithm 1 **FPML (Follow the Perturbed Multiple Leaders)** would be more appropriate, and we will make this change to the paper.

---

### Official Review · Reviewer_AB1a · 2022-07-06

**Rating:** 7
**Confidence:** 3
**Soundness:** 4 excellent
**Presentation:** 3 good
**Contribution:** 3 good

**Summary:**

The paper studies a variant of adversarial online learning which has a particular trade-off between budget and regret bounds in adversarial online learning. Standard adversarial online learning is a sequential setting. In each round $t = 1, \ldots, T$ the learner picks an action out of $N$ possible actions, suffers a loss, which is revealed to the learner afterwards. The goal is to control the regret, which the difference between the cumulative loss of the learner and the cumulative loss of the best fixed action.
In the budgeted version that is studied in this paper, the learner gets to choose a set of actions with cardinality $B$ each round and suffers the minimum loss of the actions in the chosen set. The definition of the regret does not change, which means that the learner gets more power compared to the standard version of the game.

First, the authors analyse the full feedback setting and present a new algorithm, \textit{Follow the Top Perturbed Leaders}, and its regret upper bound, which is $O(T^{\frac{1}{1+B}}\ln(N)^{\frac{B}{1+B}})$. Then, they adapt this algorithm and its upper bound to the semi-bandit feedback setting.

The new algorithm is applied as a subroutine of an online greedy algorithm in Online Submodular Function Maximization. Interestingly, in this application the authors show that one can also prove bounds for a stronger notion of regret, namely against the best fixed set rather than against the best fixed action. Finally, they present experiments for online hyperparameter optimization problems in the partial feedback setting.

**Questions:**

What type of estimators should be used for the partial information setting and what type of regret bounds do such estimators give?

**Limitations:**

The authors have adequately addressed the limitations and potential negative societal impact of their work.

**Strengths And Weaknesses:**

Strenghts

    - The setting is well motivated. The list of possible applications in the introduction is extensive and the application to submodular optimization is elegant. The applications provided by the authors in section 5 of the paper are also relevant.
    - The authors study the trade-off between budget and regret bounds both in the full feedback and semi-bandit feedback settings. For both settings, they provide the analysis of regret upper bounds, which clearly show how the trade off between budget and regret works in different settings.
    - The analysis of the algorithm is a natural extension of the analysis of follow the perturbed leader and is well described. The analysis appears to be technically sound, although I have one minor comment, see below.
    - The paper is well written given space constraints, although it can be a bit dense at times seemingly due to these space constraints. Particularly the section on submodular optimization was difficult to read because it seemed to be quite compressed.
    - The theory is supported by providing experiments, both on a known public benchmark  and on synthetic data. The experiments even revealed some unexpected behaviour of the algorithms, leading to an interesting future research question.

Weaknesses

    - For me the main weakness of the paper is the partial information bound. While the generality is nice, it also does not seem to give the optimal rate in the standard bandit setting ($B=1$) and the best I can get is a $O(T^{2/3})$ bound, where I hide dependency on $N$. This also makes me believe that the approach does not give satisfactory results in the cases where $B>1$, although I do not have a point of reference for these cases.
    - The authors show a lower bound only for the full feedback setting and as the authors acknowledge this result is weak. It would have been nice to have a stronger one, to better understand the nature of the problem.
    - The authors provide a regret bound for \textit{OG} which uses \textit{Follow the Top Perturbed Leaders} as a subroutine in the full feedback setting, but they do not for the partial feedback setting. They do give a hint for the adaptation to the partial feedback setting and test its performance in the experiments, but it would have been a more complete work if they also provided the bound.

Minor comments

    - In lines 156 to 165 I have trouble understanding the role of $S_t$ versus $S_t^*$. I think $S_t$ should be $S_t^*$, or can they be used interchangeably?
    - Typo for the first character at line 363.
    - The acronym for follow the top perturbed leader is a bit unfortunate as in literature FTL usually means Follow the Leader. Perhaps a change of acronym is in order.
    - In line 160, it seems that the second strict inequality should be an equality

---

> ### Author Response · Authors · 2022-07-31
> **Response**
>
> We thank the reviewer for their valuable time and helpful comments, which we address individually below.
>
> > For me the main weakness of the paper is the partial information bound. While the generality is nice, it also does not seem to give the optimal rate in the standard bandit setting ($B=1$) and the best I can get is a $O(T^{2/3})$ bound, where I hide dependency on $N$.
>
> Perhaps the reviewer could clarify which bound they are referring to? It seems most likely to us that Proposition 5 is the one in mind, but we apologize if we have misunderstood. This result states that the algorithm's regret in the partial feedback setting is bounded by $\ln(N)/\epsilon + T(1-\mathrm{e}^{-K\epsilon})^B$ if the cost estimates are bounded by $K$. When choosing $\epsilon = (\ln(N)/TK^B)^{1/(B+1)}$ this regret bound is asymptotically $\mathcal{O}(T^{1/(B+1)}(K\ln(N))^{B/(B+1)})$, which in the standard bandit setting ($B=1$) reduces to $\mathcal{O}(\sqrt{T K \ln(N)})$.
>
> This behaviour is essentially optimal in the standard bandit setting, as far as we are aware: the Exp3 algorithm experiences regret bounded by $\mathcal{O}(\sqrt{TN\ln(N)})$, for instance, which is known to be optimal in $T$ (e.g. Auer et al. [1995]) and which our algorithm matches when $K = \mathcal{O}(N)$ (a weak condition). We are not sure where the reviewer's $T^{2/3}$ came from but would gladly help if they could elaborate.
>
> > The authors provide a regret bound for $\textit{OG}$ which uses $\textit{Follow the Top Perturbed Leaders}$ as a subroutine in the full feedback setting, but they do not for the partial feedback setting. They do give a hint for the adaptation to the partial feedback setting and test its performance in the experiments, but it would have been a more complete work if they also provided the bound.
>
> We thank the reviewer for this helpful point. We decided to omit the regret bound for $\mathbf{OG_{hybrid}}$ in the partial feedback setting due to lack of space and not being the main focus of the paper, but as alluded to at the end of Section 3 such a bound is actually obtainable through a trivial proof adaptation; the bound in Theorem 6 simply becomes $\mathcal{O}(B'\ln(T)T^{1/(\tilde{B}+1)}(K\ln(N))^{\tilde{B}/(\tilde{B}+1)})$ where $K$ is the bound on the cost estimates used by the subroutine, reducing to $\mathcal{O}(B'\ln(T) N\ln(N))$ in the special case where $\tilde{B} = \lceil \ln(T) \rceil$ if $K = \mathcal{O}(N)$. We are happy to mention this bound in the final version.
>
> > What type of estimators should be used for the partial information setting and what type of regret bounds do such estimators give?
>
> We thank the reviewer for this astute question. There is a substantial discussion to be had here, which would be beyond the scope of the paper, but a good choice would ideally be estimators of the form
> $$\hat{c}_t(a) := \begin{cases} \frac{c_t(a)}{\mathbb{P}(a \in S_t)} & \text{if } a \in S_t, \\\\ 0 & \text{otherwise} \end{cases}$$
> where $S_t$ is the set of arms chosen by the algorithm at round $t$ and $\mathbb{P}$ refers to the distribution over arms implied by the algorithm at round $t$. This ensures the estimators are unbiased as required (and also exhibits a self-stabilizing character that makes additional explicit exploration unnecessary, since an arm not pulled for a few rounds will have a lower cumulative estimated cost than others and so will be more likely to be pulled again soon).
>
> The problem is that the probability $\mathbb{P}(a \in S_t)$ does not have a closed form and is expensive to calculate numerically. One solution inspired by Neu and Bartok [2016] is to re-run the algorithm again repeatedly at round $t$ (after making the choice $S_t$) until every arm $a \in S_t$ has been pulled again by the algorithm at least once. The number of repetitions $M_a$ necessary until arm $a$ is pulled again has distribution $\mathrm{Geom}(\mathbb{P}(a \in S_t))$ and so is an unbiased estimate of $1/\mathbb{P}(a \in S_t)$ and can be used in its place in the cost estimate formula above.
>
> In practice, this leads to potentially large sampling times so it is useful to cap the number of repetitions allowed by some $K$, leading to estimators
> $$\hat{c}_t(a) := \begin{cases} c_t(a) \mathrm{min}(M_a,K) & \text{if } a \in S_t, \\\\ 0 & \text{otherwise}. \end{cases}$$
> This introduces a small bias but greatly reduces the variance in $\hat{c}_t(a)$, which is now bounded above by $K$, and a simple adaptation to the proof of Proposition 5 shows that this leads to an attractive regret bound. These *geometric resampling* estimators are the ones that we use in our experiments, and we touch on this formally in Appendix C.3 of the paper.
>
> ### $S_t$ vs $S_t^*$ in L156–165
>
> The reviewer is correct that these refer to the same object; we will fix the asterisk omission!
>
> ### L363 typo; L160 inequality
>
> The reviewer is correct on both fronts; thanks.
>
> ### FTL acronym
>
> We agree and will change the name to **FPML (Follow the Perturbed Multiple Leaders)**.

---

> > ### Comment · Reviewer_AB1a · 2022-08-08
> > **Response**
> >
> > Thank you for the detailed response.
> >
> > I was indeed referring to proposition 5. In my response I will restrict to the bandit setting. I will use $a_t$ to denote the action in round $t$ and throughout I will assume that the cost in bounded by 1. In proposition 5 $K$ is the upper bound on the loss estimates. In your response, you claim that $K = O(N)$ is a weak assumption, where $N$ is the number of actions. Let me explain why this is in fact not a weak assumption.
> >
> > Suppose we use the standard importance weighted estimator. Then the importance weighted estimator is given by
> > $$
> > \hat{c}_t(a) = \frac{1_t(a) c_t(a)}{P_t(a_t = a)}
> > $$
> > where $1_t(a)$ is the indicator function for $a = a_t$ and $P_t(a_t = a)$ is the probability that the action in round $t$ is equal to $a$. If we assume that $\hat{c}_t(a)$ is bounded by $\frac{N}{\gamma}$ for some $\gamma \in (0, 1]$, then this implies that $P_t(a_t = a) \geq \frac{\gamma}{N}$ for all $t$ and $a$. Suppose that $c_t(a) = 1$ for all suboptimal actions and $c_t(a) = 0$ for the optimal action. This implies that in any given round we suffer at least $\gamma\frac{(N-1)}{N}$ instanteneous regret, as we put at least $\gamma \frac{N-1}{N}$ of the total probability distribution on suboptimal actions. Summing the instanteneous regret over the rounds, we can see that the total regret is at least $\gamma \frac{N-1}{N}T$. Thus if we set $\gamma$ to be any constant independent of $T$ as we do when assuming $K = O(N)$, we will suffer linear regret!
> >
> > For the geometric resampling estimator you refer to a similar problem occurs. Specifically, if we look at equations (8) and (9) in Neu and Bartok (2016) we see that the bias in any given round is $O(\frac{1}{K})$. Thus, the total bias after $T$ rounds is $O(\frac{T}{K})$. I believe we do not have any control of $M_a$, therefore the best bound I can find is
> > $$
> > O(\sqrt{KTln(N)} + \frac{T}{K}).
> > $$
> > Setting $K = T^{1/3}$ gives us $\tilde{O}(T^{2/3})$ regret. However, setting $K = N$ again leads to linear regret.
> >
> > Neu, G., & Bartok, G. (2016). Importance weighting without importance weights: An efficient algorithm for combinatorial semi-bandits.

---

> > > ### Author Response · Authors · 2022-08-09
> > > **Response**
> > >
> > > Thank for your response and very clear explanation. We agree with you that in the $B=1$ case the $K = \mathcal{O}(N)$ condition is in fact not likely, and that the geometric resampling estimators do lead to a $\mathcal{O}(T^{2/3})$ bound using our analysis when $B=1$. Interestingly, in this case our algorithm is equivalent to the algorithm from Neu and Bartok (2016), for which they proved an $\mathcal{O}(\sqrt{T})$ bound; this leads us to believe that a tightening of our analysis (possibly by considering per-round cost estimate bounds rather than a single global upper bound) may lead to an improved result, and this would be an interesting direction for future work. Thank you for highlighting this.

---

### Official Review · Reviewer_Gn74 · 2022-07-11

**Rating:** 6
**Confidence:** 3
**Soundness:** 3 good
**Presentation:** 2 fair
**Contribution:** 3 good

**Summary:**

This paper considers a new learning with experts' advice setting where the learner could choose B (B\geq 1) experts and suffers a minimal loss of the chosen exprets. The authors propose an extension of the classical FTPL algorithm for solving the problem, and the basic idea is to choose the top-K perturbed leader at each round. The authors show that the algorithm could trade-off between regret and B, which could reduces to minimax results when B=1 or N. Finally, the authors consider several extensions, including bandit feedback, OSFM, and experiments on real-world datasets.


**Questions:**

I’m not an expert in this area and I was wondering if the authors to clear some of my concerns: 1) what is the exact relationship between the problem considered in this paper and OSFM, and are there any existing methods can solve this problem? 2) Can the authors compare the regrets between OG and OG_hybrid?

I think the authors should also cite the line of work about online submodular minimization, such as [Hazan & Kale., 2012].

Hazan, E., & Kale, S. (2012). Online Submodular Minimization. Journal of Machine Learning Research, 13(10).

**Ethics Review Area:**

["I don’t know"]

**Limitations:**

Yes

**Strengths And Weaknesses:**

Strengths:

The proposed setting has a strong relationship with online submodular minimization, and online combinatorial minimization, but to my knowledge the introducing of budget is new and novel.

The proposed algorithm is a direct extension of the classical FTPL algorithm, and I am bit surprised to see that this version of FTPL (choosing the top K leaders) is not considered by any previous work. Although the proof is not so complicated, I think the idea is novel and the theoretical guarantees on the regret-budget trade-off is also interesting and novel.

The authors did a solid work and considered several extensions, including the bandit case, application in OSFM, and the also did several experiments to demonstrate the effectiveness of their methods.


Weakness:

It would be great if the authors could get a B related lower bound.

Presentation: in the introduction, the authors give a clear description of the OLwE with budget problem, and also the corresponding Top-K FTPL algorithm. However, the part related to OSFM (start from Line 60) is not easy to read, and I think the readers may have the several  questions, including what is the exact relationship between OLwE with budget and OSFM, and intuitively, What is “best fixed schedule of length B”, and what is the improvement of the proposed algorithm compared to OG in terms of the order of regret (something related to B)?
I think a rewrite of Lines 60-65 may make the introduction easier to read.

---

> ### Author Response · Authors · 2022-07-31
> **Response**
>
> We thank the reviewer for their constructive comments and taking the time to review this paper. We also thank the reviewer for their suggestion to make the relationship between OSFM more explicit in the introduction. We agree this would be helpful, and we will make the suggested changes.
>
> > What is the exact relationship between the problem considered in this paper and OSFM, and are there any existing methods can solve this problem?
>
> (For clarity: we believe the reviewer may have intended to write Online Submodular Function *Maximization* in their review, which is the problem considered in Section 3.)
>
> The problem first introduced in this paper can be considered as a special case of OSFM. As discussed in L84, if we consider working with rewards $r_{t}(a)=1-c_{t}(a)$ instead of costs, then the problem of the paper is to give an algorithm which chooses action sets $S_t$ such that
> $$R_{T}=\max_{r_1,\dots,r_t} \mathop{\mathbb{E}}\left[\max_{a \in \mathcal{A}} \sum_{t=1}^T r_{t}(a)-\sum_{t=1}^{T} \max_{a \in S_t} r_{t}(a)\right]$$
> is small (formally, it is easy to show that a bound on $R_{T}$ in this max-of-rewards formulation automatically implies a bound on $R_{T}$ in the min-of-costs formulation).
>
> Because $\max$ is submodular, if we define the submodular function $f_{t}(S_t)=\max_{a \in S_t} r_{t}(a)$, then we have
> $$R_{T}=\max_{r_1,\dots,r_t} \mathop{\mathbb{E}}\left[\max_{a \in \mathcal{A}} \sum_{t=1}^T f_{t}(a)-\sum_{t=1}^{T} f_t(S_t)\right]=\max_{r_1,\dots,r_t} \mathop{\mathbb{E}}\left[\mathrm{OPT}(1)-\sum_{T=1}^{T} f_t(S_t)\right],$$
> where $\mathrm{OPT}(k):=\max_{\substack{S \subset \mathcal{A}\\\\|S|=k}} \sum_{t=1}^{T} f_{t}(S)$ is the score of the best fixed subset of $k$ arms in hindsight. Our problem can therefore be viewed as a special case of online submodular function maximization where (a) the submodular function is $\max$ and (b) we are trying to minimize regret against $\mathrm{OPT}(1)$. Prior work could not solve this problem because it only gives bounds against $(1-e^{-1})\mathrm{OPT}(B)$, not $\mathrm{OPT}(1)$, where $B$ is the budget of the algorithm.
>
> In Section 3 of the paper, it is shown that solving the above problem (i.e. this special case of OSFM) is enough to give a solution to the general problem of OSFM (i.e. any submodular reward function) against a regret benchmark $\mathrm{OPT}(B')$ which is independent from the budget $B$ (Theorem 6). This new result allows us to quantify how performance increases as the budget increases while holding the benchmark budget $B'$ fixed, which was not possible with the bound from prior work.
>
> > Are there any existing methods can solve this problem? Can the authors compare the regrets between OG and OG_hybrid?
>
> We are unaware of any prior work which solves the bandits problem proposed in the paper for $B>1$. In the case of OSFM, we are unaware of any prior techniques which are designed to give improved bounds on $R_{T}=\max_{f_1,\dots,f_t} \mathop{\mathbb{E}}\left[\mathrm{OPT}(B')-\sum_{T=1}^{T} f_t(S_t)\right]$ when $B'<B$. The algorithm $\mathbf{OG}(B)$ given in Streeter and Golovin [2008] (L226) for OSFM gives the guarantee
> $$R_{T}=\max_{f_1,\dots,f_t} \mathop{\mathbb{E}}\left[(1-e^{-1})\mathrm{OPT}(B)-\sum_{t=1}^{T} f(S_t)\right] \leq \mathcal{O}(\sqrt{TB \ln(N)})$$
> when the algorithm has a budget of $B$. To give some comparisons:
>
> 1. $\mathbf{OG_{hybrid}}(B,1)$ is the same as $\mathbf{OG}(B)$, and so it has the same regret bound as $\mathbf{OG}(B)$ given above.
>
> 2. $\mathbf{OG_{hybrid}}(B(\ln T)^2,\ln T)$ has a budget of $B(\ln T)^2$ arms per round, and has a regret bound (from the final conclusion of Theorem 6) of
>    $$R_{T}=\max_{f_1,\dots,f_t} \mathop{\mathbb{E}}\left[\mathrm{OPT}(B)-\sum_{t=1}^{T} f(S_t)\right] \leq \mathcal{O}(B\ln(T)\ln(N)).$$
>    In the case of online hyperparameter optimization, the interpretation is that if you are willing to increase your runtime (budget) by a factor of $(\ln T)^2$, you can improve your hyperparameter optimizer score guarantee by a factor of $(1-e^{-1})$ and have the additive regret grow like like $\mathcal{O}(\ln(T))$ instead of $\mathcal{O}(\sqrt{T})$. We are therefore able to quantify a trade-off between budget constraints and regret bounds.
>
> 3. $\mathbf{OG}(B\ln(T)^2)$ has a budget of $B\ln(T)^2$ arms per round, and gives the guarantee  $$\mathop{\mathbb{E}}\left[(1-e^{-1})\mathrm{OPT}(B\ln(T)^2)-\sum_{t=1}^{T} f(S_t)\right] \leq \mathcal{O}(\sqrt{T\ln(T)^2 B \ln(N)})$$. If we compare this with (2) above, $\mathbf{OG}(B\ln(T)^2)$ competes with the benchmark $(1-e^{-1})\mathrm{OPT}(B\ln(T)^2)$ instead of $\mathrm{OPT}(B)$, which is not immediately comparable. The additive regret of $\mathbf{OG_{hybrid}}(B(\ln T)^2,\ln T)$ has a smaller dependence on $T$ and a larger dependence on $B$ than the additive regret bound of $\mathbf{OG}(B \ln(T)^2)$. Unlike in (2), it is not easy to quantify the relative improvement between $\mathbf{OG}(B)$ and $\mathbf{OG}(B\ln(T)^2)$ because the benchmark changes.

---

### Official Review · Reviewer_2itn · 2022-07-11

**Rating:** 4
**Confidence:** 3
**Soundness:** 2 fair
**Presentation:** 2 fair
**Contribution:** 2 fair

**Summary:**

They propose a new online learning setting which is in each round $[S_t]$ arms are pulled but the learning algorithm only suffers the regret of the arm with the lowest cost among all the pulled arms. They propose a new learning algorithm, Follow the Perturbed Leader (FPL), for it. In addition, both theoretical and practical results are provided.



**Questions:**

see above

**Ethics Review Area:**

["I don’t know"]

**Limitations:**

see above

**Strengths And Weaknesses:**

Weakness: Comments on the writing:
There is no proper section stating the learning problem. Also, I do not understand the feedback model such as whether all arms in $[S_t]$ will be observed or not, based on their Algorithm~1.

Based on my understanding, if all arms in $[S_t]$ are observed at the end of each round, it is more like graphical bandit setting that is shown in (From Bandits to Experts: On the Value of Side-Observations, Shie Mannor, Ohad Shamir, 2011).

Also, I believe adversarial online learning are quite understood. So, I do not think the proposed learning problem is challenging.

---

> ### Author Response · Authors · 2022-07-31
> **Response**
>
> We thank this reviewer for their comments. We hope our responses below may help and perhaps convince the reviewer to raise their score.
>
> > Comments on the writing: There is no proper section stating the learning problem.
>
> We believe we have stated the learning problem in sufficient formal detail at the start of Section 1 (see in particular the first paragraph and L43–48), but if the reviewer would like to suggest any improvements to this exposition we would of course be happy to consider them.
>
> > Also, I do not understand the feedback model such as whether all arms in $[S_t]$ will be observed or not, based on their Algorithm~1. Based on my understanding, if all arms in $[S_t]$ are observed at the end of each round, it is more like graphical bandit setting that is shown in (From Bandits to Experts: On the Value of Side-Observations, Shie Mannor, Ohad Shamir, 2011).
>
> In the paper we consider first the full feedback setting, where the algorithm observes the full cost function $c_t$ after making its choice and then a generalization to the *semi-bandit feedback* setting where the algorithm observes the cost of each arm it pulled. Again, we apologize for any confusion; we believe that we have been clear in the text about this (see L21–22, 122, 127, 197–209) but would of course be open to any specific suggestions.
>
> Concerning the citation mentioned, we believe there may have been a slight confusion in the reviewer's understanding of the learning problem we are considering. While interesting, the work mentioned seems to us to consider a problem quite unrelated to ours. Theirs involves choosing a single action before receiving side observations about some others, inducing a graph-like structure; in ours we choose a size-constrained set of actions and receive the minimum cost of all the chosen actions, and there is no side information (in the partial feedback setting). The structure and nature of the problems do not appear closely related to us, but if the reviewer would like to expand on their comparison we'd be happy to discuss this more.
>
> > Also, I believe adversarial online learning are quite understood. So, I do not think the proposed learning problem is challenging.
>
> While the case of linear costs is well-understood by the combinatorial adversarial bandit literature, the case of non-linear costs, applicable to our learning problem, is comparatively less studied; see Section 1.1 for a full discussion. Moreover, the specific form of regret we consider in our paper has not to the best of our knowledge been studied before.

---

### Meta-Review · Area_Chair_gLTS · 2022-09-03

**Recommendation:** Accept
**Confidence:** Less certain

**Metareview:**

The paper considers the experts problem when the online player is allowed to choose B arms instead of 1 and is rewarded according the best arm within the set it chose. The comparator considered in the first result of the paper is standard best single arm. The paper shows that the regret scales in this setting as ~ T^(1/(B+1)) -- recovering and extending the standard result. The paper uses this result to then further get another result when the set wise function is max. Here existing result in the comparator compared with fixed size B, i.e. max size allowed, where as they present a result that allows comparison with any budget B'. They also play a ln(T) factor more arms in each round which allows them to compare with OPT as opposed to (1 - 1/e) OPT.

The paper according to reviews is well written, but I found section 3 hard to read, especially the notations of B' and tilde{B}. I also think the regret statement in that theorem is not well defined, you should state clearly it is with respect to B' sized sets. There are other shortcomings of the paper like the bandit results dont match the optimal in the case of B=1 (as discussed with a reviewer), which points to suboptimality of presented bound. The lower bounds presented are also unfortunately weaker. These are not shortcomings per se but desiderata to make the paper very strong. I also find the fact that they ln(T) factor more arms in OSFM is not stressed appropriately.

Overall the results in the paper are good as presented, but the shortcomings listed above the paper put it right on the borderline. Overall there is general appreciation of the results by the reviewers and as such according to that it puts it slightly above the borderline according to me. I am recommend a marginal accpet for the paper. I would like to urge the authors to look at the paper critically irrespective of the outcome and definitely look to improve presentation at the minimum. Other lingering questions if answered will be a great bonus.

**Award:**

No

---

### Decision · Program_Chairs · 2022-09-14

Accept